# Why Has Metabolomics So Far Not Managed to Efficiently Contribute to the Improvement of Assisted Reproduction Outcomes? The Answer through a Review of the Best Available Current Evidence

**DOI:** 10.3390/diagnostics11091602

**Published:** 2021-09-02

**Authors:** Charalampos Siristatidis, Konstantinos Dafopoulos, Michail Papapanou, Sofoklis Stavros, Abraham Pouliakis, Anna Eleftheriades, Tatiana Sidiropoulou, Nikolaos Vlahos

**Affiliations:** 1Second Department of Obstetrics and Gynecology, “Aretaieion Hospital”, Medical School, National and Kapodistrian University of Athens, Vas. Sofias 76, 11528 Athens, Greece; mixalhspap13@gmail.com (M.P.); annielefth-28@hotmail.com (A.E.); nikosvlahos@med.uoa.gr (N.V.); 2Department of Obstetrics and Gynecology, Assisted Reproduction Unit, Faculty of Medicine, School of Health Sciences, University Hospital, University of Thessaly, 41110 Larissa, Greece; kdafop@med.uth.gr; 3First Department of Obstetrics and Gynecology, Assisted Reproduction Unit, Medical School, Alexandra Hospital, National and Kapodistrian University of Athens, 80 Vas. Sofias Av. and Lourou str., 11528 Athens, Greece; sfstavrou@yahoo.com; 4Second Department of Pathology, “Attikon” University Hospital, National and Kapodistrian University of Athens, Rimini 1, Chaidari, 12642 Athens, Greece; apouliak@med.uoa.gr; 5Second Department of Anesthesiology, Attikon University Hospital, National and Kapodistrian University of Athens, Rimini 1, 12462 Athens, Greece; tsidirop@med.uoa.gr

**Keywords:** in vitro fertilization, assisted reproductive techniques, metabolomics, biomarkers, follicular fluid, diagnosis

## Abstract

Metabolomics emerged to give clinicians the necessary information on the competence, in terms of physiology and function, of gametes, embryos, and the endometrium towards a targeted infertility treatment, namely, assisted reproduction techniques (ART). Our minireview aims to investigate the current status of the use of metabolomics in assisted reproduction, the potential flaws in its use, and to propose specific solutions towards the improvement of ART outcomes through the use of the intervention. We used published reports assessing the role of metabolomic investigation of the endometrium, oocytes, and embryos in improving clinical outcomes in women undergoing ART. We initially found that there is no evidence to support that fertility outcomes can be improved through metabolomics profiling. In contrast, it may be helpful for understanding and appraising the nutritional environment of oocytes and embryos. The causes include the different infertility populations, the difference between animals and humans, technical limitations, and the great heterogeneity in the variables employed. Suggested steps include the standardization of variables of the method itself, the universal creation of a panel where all biomarkers are stored concerning specific infertile populations with different phenotypes or etiologies, specific bioinformatics contribution, significant computing power for data processing, and importantly, properly conducted trials.

## 1. Introduction

There is a tendency for the identification and utilization of markers, with sufficient sensitivity and specificity that may lead to change of management to improve outcomes [1,2], in all aspects of medicine. These aspects may include cancer treatment, but also fertility management and assisted reproduction techniques (ART) [3,4]. As such, metabolomics emerged to give clinicians the necessary information on the competence, in terms of physiology and function, of gametes, embryos, and the endometrium towards a targeted infertility treatment, namely, assisted reproduction techniques (ART). Various metabolites that were associated with certain cellular activities were evaluated in this concept. As an example, embryo culture media metabolites such as single biomarkers (pyruvate, glucose, amino-acids, oxygen, and leptin) or endogenous metabolites of different classes (acyl carnitines; amino acids; hexose; sphingolipids; glycerophospholipids; biogenic amines; steroid hormones: mineralocorticoids, glucocorticoids, and sex steroids) were investigated so far [5,6,7]: these can be assessed and provide the necessary information for the dynamics of the embryo.

Infertility remains a global socioeconomic problem and ART were implemented towards the aim of establishing a live birth; unfortunately, the effectiveness of ART is limited, as only 10–30% of all embryos replaced in the uterus will implant and finally result in a live birth [8]. In a recent review on the effectiveness and safety of metabolomic assessment of oocyte quality, embryo viability, and endometrial receptivity for improving live birth in women undergoing ART compared to that of conventional methods of assessment, authors concluded that there is no evidence to show that metabolomic assessment of embryos before implantation has any meaningful effect on success rates, while the existing evidence ranged from very low- to low-quality [9].

In contrast, there is ongoing reporting of studies towards this goal, but only at the diagnostic level; some fine examples include the use of metabolomics in the follicular fluid of women undergoing in vitro fertilization (IVF), towards the understanding of the environment of the oocyte development [10,11,12,13,14,15,16,17] or of the embryo competency [18,19,20], in obese patients intending to undergo IVF [21] or for those with diminished ovarian reserve [22], in maternal plasma in the late first trimester of pregnancy [11], and in human sperm and seminal plasma [23].

The rationale of this paper is based on the fact that there is ongoing research activity on metabolomics, but this did not manage to improve the success rates in assisted reproduction so far. Our minireview therefore aims to investigate the utilization of metabolomics in assisted reproduction and the potential flaws in its use, and to propose evidence-based recommendations towards the improvement of IVF outcomes.

## 2. Materials and Methods

We performed a comprehensive literature search in PubMed. Published reports of all study types assessing the role of metabolomics of the endometrium, oocytes, and embryos in improving clinical outcomes in women undergoing ART (including both intra-cytoplasmic sperm injection (ICSI) or IVF) were evaluated for possible inclusion in this review. We applied no limitations on country of origin or language. We examined the references lists of all studies and relevant reviews to identify further relevant studies.

The relevant keywords were: “IVF” or “in vitro fertilization” or “ICSI” or “intracytoplasmic sperm injection” or “ET” or “Embryo” or “Embryo Transfer” or “Follicular Fluid” or “Culture-Media” and “metabolomic” or “near infrared spectroscopy” or “NIR” or “biomarkers” or “endometrial receptivity markers” or “follicular fluid profile” or “metabolic profile” or “metabolomics” or “metabonomics” or “cometabolism” or “metabolites” or “byproducts”.

## 3. Results

### 3.1. The Intervention: Description

Metabolomics refer to the newest ‘omics’ technologies and include interactions of cellular structures DNA and genes to metabolites [24]. They are a group of small-molecule, nonproteinaceous compounds, including metabolic intermediates, adenosine triphosphate, hormones, and metabolites, which are present in a biological sample [25]. Metabolomics is more informative than genomics, transcriptomics, or proteomics because they represent the final products of the cell regulatory process, and are closer to the functional phenotype [24,26]. The metabolomic profile is a terminal cellular product that is used to distinguish between a normal and a pathological state [27] to elucidate the cellular mechanisms involved, and, consequently, to ‘foresee’ the capacity of an oocyte to progress after fertilization, the fate of an artificially produced embryo, and the efficacy of endometrium to successfully accept it [28]. Interestingly, in the paper of Beliver et al. (2012), the use of omics in the field of reproduction was named “reproductomics”; concerning the embryo, they provide valuable information on the biological processes occurring at each step of embryonic development, especially with “signs of suboptimal growing conditions that can affect embryo implantation, including low rates of division, blocked cytokinesis, cytoplasmic vesicula tion, abnormal activation of the genome, gene transcription, and energy metabolism” [29].

As recently described by the first author of a Cochrane systematic review, the term ‘metabolomics’ in assisted reproduction refers to the metabolic products found in specific biological materials or media. In the endometrium, it is associated with its ability to be receptive; for the oocyte, it refers to its fertilizing functionality and capacity, and is mediated mainly through the analysis of the follicular fluid. In the embryo, the approach is usually performed through the analysis on spent media culture [8]. One of the problematic issues of the use of metabolomics—as an omic technique—is due to its dependence almost uniquely on the number of variables considered. In this context, extra data deriving from the resident microbiota and the bacterial byproducts could contribute to the high accuracy of the method [30,31].

### 3.2. The Potential Implication

Conception by ART is associated with poor success rates and an increased incidence of obstetric and perinatal complications [32]. Gamete quality, embryo quality, and endometrial receptivity are considered crucial components to the success of ART. The integration of metabolomics into ART could assist the selection of viable embryos and competent oocytes, as well as the creation of a healthy and receptive endometrium for implantation. Specifically, metabolomics could improve the associated procedures and increase the success rates of ART by reducing implantation failures, miscarriages, multiple pregnancies, ectopic pregnancies, and fetal abnormalities, thus alleviating the emotional and socioeconomic consequences that accompany them [9].

### 3.3. The Facts

There is no evidence to support that fertility outcomes can be improved through metabolomics profiling, as there is no clear difference between metabolomic and morphology assessment of the embryo in the rates of live birth or ongoing pregnancy in women undergoing ART [8]; in contrast, it may be helpful for understanding and appraising the nutritional environment of oocytes and embryos. The clinical outcomes, namely live birth and miscarriage after ART, are crucial and remain the ultimate target for a battery of studies related to metabolomics. Yang et al. confirmed that the metabolomic profile cannot effectively contribute to the change of clinical outcomes after fertility treatments, remaining only a diagnostic tool; authors employed a liquid chromatography-tandem mass spectrometry (LC-MS/MS) analysis, where the samples were analyzed using an LC-electrospray ionization (ESI)-Tandem mass spectrometry system. Specifically, authors demonstrated that dehydroepiandrosterone (DHEA) in the follicular fluid negatively correlated with the oocyte maturation rate and the high-quality embryo rate, yet no statistical significance was reached when the association between DHEA levels and clinical outcomes (i.e., biochemical and clinical pregnancy rates) was examined [10]. Similarly, no statistically significant differences in pregnancy outcomes after ART were reported when high-density lipoprotein (HDL) levels of the follicular fluid were compared between obese and normal weighted women; authors used a fluorometric biochemical cell-free assay based on oxidation of the fluorogenic probe dihydrorhodamine 123 to assess HDL function [21]. In addition, Huo et al., using a Waters Empower 2 chromatography software and 18 individual amino acid standards, did not manage to correlate amino acid metabolomic profile during human embryo development with pregnancy outcomes after IVF [19]. Finally, in a recent study conducted by Inoue and colleagues, using a gas chromatography-mass spectrometer, the analysis of the 187 identified organic metabolites obtained by the culture medium of a single human embryo produced by IVF did not lead to improvement of pregnancy outcomes [33].

Further examples of the use of metabolomics as potential diagnostic tools are: (i) the novel high-coverage targeted metabolomics method (SWATH to MRM), used for exploring the follicular fluid metabolome alterations in women with recurrent spontaneous abortion undergoing IVF, where a total of 18 FF metabolites were identified [34], and (ii) a maternal metabolomic profile through the analysis of 17-β-estradiol and progesterone levels in maternal plasma in the late first trimester between spontaneous pregnancies and pregnancies conceived with fertility treatments; in this study, authors used two separate reverse phase/ultraperformance liquid chromatography—tandem mass spectrometry (UPLC-MS/MS) methods, quantifying a total of 806 known metabolites [11]. As reported from the studies referenced above, there is no direct evidence to support a potent linkage of these pathologies to the metabolomic profile. In contrast, there are currently reports to support the hypothesis that specific metabolites in embryo culture media can be correlated with the status of the embryos [35].

Table 1 demonstrates the variability between methods employed by different studies for metabolomics analysis of human samples [7,10,11,12,13,14,16,17,18,19,20,21,22,23,33,34,36].

### 3.4. Reasons

There are various reasons why metabolomics does not yet have the necessary efficiency as a technique to improve IVF outcomes. One of them includes the different infertility phenotypes that reports are dealing with. In a study of women undergoing IVF, pretreatment anti-Müllerian hormone (AMH) and antral follicle count (AFC) measurements were correlated with serum lipids, lipoprotein subclasses, and low-molecular-weight metabolites that were measured with nuclear magnetic resonance (NMR) spectroscopy; AMH was significantly associated with HDL, omega-6 and polyunsaturated fatty acids, and the amino acids isoleucine, leucine, tyrosine, and acetate, while AFC was significantly associated with alanine, glutamine, and glycine [22]. These data reveal that ovarian reserve markers can influence serum metabolomics and concurrently exert an impact on follicular fluid and embryo metabolomics. Although this is an issue requiring further research, one cannot exclude the possibility that different infertility phenotypes may constitute significant confounders on the association between metabolomics and IVF outcome. Other underlying infertility conditions, including polycystic ovary syndrome and endometriosis, may significantly influence the metabolomics of the follicular fluid, as measured by NMR [17,34]. Further confounding conditions potentially affecting the follicular microenvironment and negatively influencing the efforts towards the standardization of metabolomics’ assessment are the drugs used (e.g., for oocyte triggering), the size of the follicle, the cycle day of the aspiration, and the size and quality of the retrieved oocyte [17,37,38].

Another reason is the variation concerning the culture media components. Thus, this can constitute a significant confounder when NMR studies are performed on spent culture media. The effect of the composition of embryo culture media on the metabolic activity of the embryo is of paramount importance [39]. In the same context, the handling of the spent culture media may contribute to the great heterogeneity of the results in the relevant studies.

The differences of metabolomics between animals and humans should also be considered. For example, there are significant differences in the metabolism between humans and the bovine models that are experimentally used to provide insights in the embryo metabolism, so that evidence originating from animal studies cannot be extrapolated to explain or interact with human physiology, according to the recent paper by Asampille and colleagues [38]. In the same context, as the in-cell analysis of oocytes/embryos exposes them to invasive preparation, such as centrifugation and NMR, ethical issues may arise [40]; of note, potential long-term detrimental effects were not yet assessed.

Finally, there may be several technical issues that limit the practicality and routine utilization of NMR technology in IVF clinics, such as the time requirement for NMR analysis and the expertise needed to interpret the NMR results [38]. Massive untargeted, and secondarily, targeted metabolomics analyses may also involve a larger number of metabolites than those of clinical interest. All these factors lead to the provision of results that are less interpretable by the clinicians and at a time that cannot easily conform with the flow of daily clinical practice.

### 3.5. Data Analysis and Problems with the Diagnostic Power

Data produced from metabolomics modalities involve either the acquired spectra from the NMR, mass spectroscopy, or other similar instruments, or they involve a list of metabolites and features. Such methods are typically capable of identifying from a few tenths to a few thousands of different metabolites or metabolite features at a single run. Notably, it is currently not possible to identify the entire range of metabolites with a single method and/or a single run [41]. This is indicative of one of the main issues of metabolomics data processing; namely, the dimensionality (i.e., the number of features is higher than the number of subjects, namely fertility results in the IVF setting). The latter may be further exhibited by the number of metabolites included in public major databases. For instance, the Human Metabolome DataBase (https://hmdb.ca) [42] contains (as of 2 June 2021) 115,398 metabolites related to the human body, while the METLIN database contains over a million metabolites [43,44].

Data reduction techniques, such as unsupervised methods (for example Principal Component Analysis (PCA)) and PCA variants are options used to identify patterns and connections with the subject features [45,46]. Principal Coordinate Analysis (PCoA) is a fine example of these methods and is based on dissimilarity/similarity matrices (instead of using raw data as in the PCA approach); PCoA graphical representations are implemented to reveal the influence of the variables in the profile of metabolites [47]. It is usual that the metabolites of interest are not known in advance; thus, unsupervised techniques without any assumptions are considered the method of choice. The PCA approach can effectively replace the correlated metabolites using a combination of a smaller number of nonrelated metabolite data (the principal components), thus trying to resolve dimensionality issues. The disadvantage of this approach is that the combination of the metabolites forming the principal components does not lead to a direct link of the metabolites to the underlying causative mechanisms and the associated phenomena. Another issue is related to collinearity: this occurs when some variables and the causes behind them (for example, bacteria and genes) “behave” in the same way, and thus produce the same metabolites; thus, the isolation of the responsible variable becomes difficult. Further issues arise when the metabolite spectrum peaks exhibit low amplitude or are behind or at the “shoulder” of higher peaks, so that they are sometimes hidden. As a result, the most informative ones (as for example happens with bacteria species) cannot be properly recognized. In this context, scaling techniques, such as Pareto scaling and traditional spectrum evaluation methods, such as the Fast Fourier Transform (FFT) can offer important help to detect such peaks from the original signal [48,49].

Markedly, artificial neural networks (ANNs), and in general artificial intelligence (AI) methods, historically seem to have a good fit in the metabolomics arena [50]; nowadays, deep learning, being the forefront of ANNs and AI research, was already employed in metabolomics’ applications [51,52,53].

The use of the large-scale metabolomics requires the cosynergy of bioinformatics tools; the latter are used for data analysis, visualization, and integration [54,55]. The concept of the need of specialized additional tools for data visualization on one hand, and the integration of metabolomics data within a biological context on the other was expressed long ago [56]. Towards this goal, the necessary steps involve noise filtering, peak selection, deconvolution, and identification of peaks and their alignment. This eventually leads to the creation of a specific data matrix that can be subsequently used for statistical or any other type of processing, such as machine learning.

## 4. Suggested Steps

In terms of the technique in animals, the bacterial cell and the requirement of significant overexpression remain a stringent limit on the establishment of the link between structural and cell biology [40]. Nevertheless, there are significant differences in the metabolism between bovine and humans, such as in the metabolic requirements, chromatin architecture, and developmental timelines [38]. This renders the direct association of the animal studies’ results with the human metabolomic profile difficult.

Furthermore, a crucial step is the standardization of variables in terms of the method itself, either used as a complementary or independent tool for embryo selection or oocyte /endometrium assessment in ART. The above reported technical limitations for both the procedure itself and collaboration between ART clinics and NMR facilities, as well as the ability of clinicians to interpret the NMR results, are two essential points for improvement.

Properly conducted trials (namely randomized controlled trials (RCTs)) involving specific infertile target groups, encompassing live birth and miscarriage rates as their primary outcomes, are needed. In addition, these must be performed with the appropriate sample size calculated in advance of their start and free of various biases, such as selection and performance. Finally, the possibility of short- or long-term harmful effects on the tissue involved when NMR, for example, is applied should be assessed, since it may have a detrimental effect. In this respect, further studies are required for the assessment of the endometrium. The unique existing data about endometrial metabolomics analysis were focused on lipidomic analysis of endometrial receptivity. To date, they are only available in mouse models [24].

Through the discovery of new biomarkers, molecular paths could be more efficiently assessed. The complexity of the phenomena concerning these paths consists of an important barrier that can be bypassed through the collection of large amounts of data along with significant computing power for data processing. A step towards that direction could be a panel—preferably international—in which all biomarkers that are discovered by NMR and that concern specific infertile populations with different phenotypes or etiologies, such as those with repeated implantation failures or poor/high response to ovarian stimulation, are stored. A recent study protocol proposed how the measurement of different classes of metabolomics parameters may be incorporated into such a large database of patient demographics, previous cycle characteristics, used protocols, underlying infertility conditions, sperm as well as other “omics” parameters and, with the assistance of ANNs, provide valuable predictions of IVF outcomes (i.e., clinical pregnancy rates, live birth rates, miscarriage rates, multiple pregnancy rates) [53]. Even without the implementation of AI, such a database would be extremely meaningful as it would be able to address one of the main pitfalls of metabolomics’ utilization in IVF so far; that is, the inconsistency between different studies, patients, and settings [9], or even the variability between metabolomics biomarkers used [10,11,19,21,33]. Multivariable analyses conducted through the collected evidence could account for a variety of parameters and sources of heterogeneity between different infertility groups and involved IVF clinics and potentially identify specific patient subgroups in which specific metabolomics measurements lead to more accurate predictions and true improvement of IVF outcomes. Based on the implications deriving from such a database, future necessary RCTs would be designed more consistently, enrolling more homogeneous patient subsets, targeting at the enhancement of specific IVF outcomes, and finally providing high-quality data that will elucidate the true utility of metabolomics in IVF, if any. Hence, a panel of this caliber would allow the more accurate utilization of metabolomics as a tool in IVF (with that accuracy likely increasing its cost-effectiveness in clinical practice), facilitate bioinformaticians to provide more interpretable results (i.e., specific metabolic footprints) and at a more affordable time, increase our comprehension of metabolic profiles of the embryo, oocyte or endometrium during implantation (as these profiles could be associated with a large quantity of raw patient data), and, concurrently, pave the way for a more personalized approach towards IVF patients. All these concepts are summarized in Figure 1.

The standardization of the technique for the oocyte and the endometrial assessment in ART, the cooperation of various IVF Units (both in private and university hospitals), the conduction of RCTs with proper sample size power calculation, the study of infertile patients with repeated implantation failures or poor/high response to ovarian stimulation, and the involvement of AI are the next aims of the current investigation group.

## 5. Conclusions

In conclusion, current very low- to low-quality evidence failed to demonstrate any significant efficacy of metabolomics in improving ART outcomes. Such an effect may be chiefly attributed to the variability between metabolomics biomarkers measured by existing studies, as well as the substantial heterogeneity between their investigated populations and involved settings; the latter particularly regards the variability between technical parameters or even used drugs and protocols. Other factors potentially involved include the discrepancies between bovine models and humans, the time required for NMR analyses, and the technical issues arising when attempting to establish collaborations between NMR facilities and IVF Units. However, ground still exists for the use of metabolomics in ART, in case of standardization of the method itself and then incorporation into a large international database of patients in which demographics, previous cycle characteristics, underlying infertility etiologies, used protocols and drugs, sperm parameters, or even other “omics” parameters are also recorded. Analyses conducted through such a platform could account for various sources of clinical heterogeneity and identify specific subsets of patients or settings for which specific metabolomics biomarkers may be of higher benefit. In this way, randomized trials with lower inconsistency would be designed, thus allowing the elucidation of metabolomics’ role in ART, if any. For such a scenario to be feasible, the bioinformatics’ contribution as well as the establishment of a more solid collaboration between IVF clinics and NMR facilities is mandatory.

## Figures and Tables

**Figure 1 diagnostics-11-01602-f001:**
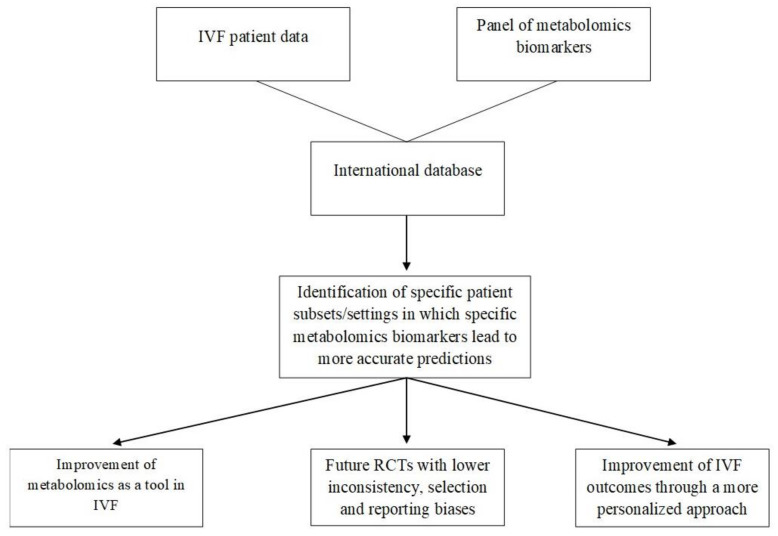
A summary of how a large database of metabolomics biomarkers measured in different IVF populations could potentially elucidate role of metabolomics in IVF and improve IVF outcomes. Abbreviations: IVF, in vitro fertilization, RCTs, randomized controlled trials.

**Table 1 diagnostics-11-01602-t001:** Demonstrates variability between methods used for metabolomics analysis of human samples. All studies included in table are original studies with human participants.

Metabolomics Method	N	Study	Study Design	Method Details Per Study	Analyzed Samples
LC-MS	8	Yang et al., 2020 [10]	Observational Cohort	LC-tandem MS	FF
		Sun et al., 2019 [11]	Observational Cohort	UP LC-MS with positive ion-mode ESI/negative ion-mode ESI/hydrophilic-interaction chromatography	Plasma
		Sun et al., 2018 [12]	Observational Cohort	UP LC with to time-of-flight MS using SWATH mode	FF
		Zhang et al., 2020 [13]	Observational Cohort	UP LC with high-resolution MS using SWATH mode	FF
		Huo et al., 2020 [19]	Observational Cohort	High performance LC	ECM
		Chen et al., 2016 [20]	Observational Cohort	Nanoscale LC coupled to tandem MS	FF
		Engel et al., 2019 [23]	Observational Cohort	LC-MS	Sperm, seminal plasma
		Song et al., 2019 [34]	Observational Case-control	UP LC with high-resolution MS using SWATH to MRM mode	FF
GC-MS	2	Ruebel et al., 2019 [14]	Observational Cohort	GC quadrupole time-of-flight MS & charged-surface hybrid column-ESI quadrupole time-of-flight tandem MS	FF, serum
		Inoue et al., 2021 [33]	Observational Cohort	GC-MS	ECM
NMR Spectroscopy	3	Castiglione Morelli et al., 2020 [17]	Observational Cohort	^1^H NMR spectroscopy	FF
		Al Rashid et al., 2020 [22]	Observational Cross-sectional	NMR Spectroscopy	Serum
		Karaer et al., 2019 [36]	Observational Case-control	NMR Spectroscopy	FF
NIR Spectroscopy	1	Vergouw et al., 2012 [7]	RCT	NIR Spectroscopy	ECM
Raman spectroscopy	1	Liang et al., 2019 [18]	Observational Cohort	Raman spectroscopy	ECM
Fluorometric measurement by CL	1	Nasiri et al., 2019 [16]	Observational Cross-sectional	Fluorometric measurement by CL system & glucose assay kit	ECM
Fluorometric biochemical cell-free assay	1	Bacchetti et al., 2019 [21]	Observational Cohort	Fluorometric biochemical cell-free assay based on oxidation of the fluorogenic probe dihydrorhodamine 123	FF

Abbreviations: N, number of studies; LC, liquid chromatography; MS, mass spectrometry; UP, ultra-performance; ESI, electrospray ionization; GC, gas chromatography; CL, chemiluminecence; FF, follicular fluid; ECM, embryo culture media; NIR, near-infrared spectroscopy; RCT, randomized controlled trial; NMR, nuclear magnetic resonance; ^1^H, hydrogen-1.

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
