# Peer review of "Why Has Metabolomics So Far Not Managed to Efficiently Contribute to the Improvement of Assisted Reproduction Outcomes? The Answer through a Review of the Best Available Current Evidence"

_diagnostics, 2021, doi:10.3390/diagnostics11091602_

Round 1
Reviewer 1 Report
The review by Siristatidis et al entitled “why has metabolomics so far not managed to efficiently contribute to the improvement of assisted reproduction outcomes?” analyse the last publication in metabolomics and the results obtained in gamete quality, embryo selection and endometrial receptivity status.
This paper is interesting but this reviewer suggest some points to be addressed:
Major point.
- The definition of Metabolomics in line 90 is too simple and not concrete, in my humble opinion. In fact they don’t say mention anything about the size of the molecules, something very relevant to discriminate a metabolite. It would be interesting to include the term “Reproductomics” in line 103 as the global omics in reproduction. This word was coined by Bellver et al., 2012 in Expert Review Obstetrics and Gynecoloy.
- The mentioned paper such as Yan et al., (Line 124) and most of them, the author talks about the discoveries without mentioning the techniques used by the researchers.
- They don’t mention either the number of metabolites they analyzed in their papers.
It is important for this reviewer to understand what techniques are used by the reviewed papers and what they analyze to consider the relevance of the discoveries they publish.
- Line 187. There are machines such as quadrupole that analyzes thousands of metabolites. It would be interesting to present a table with techniques and their limitations in detection (resolution). It would be also important to have a list of method of analysis and their utility (PCA, ANNs, etc).
- Finally, when they say that there is no evidence that the metabolomics can not be used for embryo selection they are not taken into account several abstracts and oral presentations of Cabello et al., at ESHRE and ASRM congress at the last two years. I know that they are publication in congresses but I believe that they discoveries are very promising and could be included in the review.
Author Response
This paper is interesting but this reviewer suggest some points to be addressed:
A: The authors are grateful for the time and effort in reviewing this manuscript and the constructive comments aiming to improve the overall performance of this manuscript. Please kindly note that revisions are highlighted using the “Track Changes” function throughout the text. Moreover, you will find the reference list altered. Herein follows the point-by-point response.
Major point.
- The definition of Metabolomics in line 90 is too simple and not concrete, in my humble opinion. In fact they don’t say mention anything about the size of the molecules, something very relevant to discriminate a metabolite. It would be interesting to include the term “Reproductomics” in line 103 as the global omics in reproduction. This word was coined by Bellver et al., 2012 in Expert Review Obstetrics and Gynecoloy.
A: Our goal is, at that section, to provide with a brief definition, but we have added supplementary data according to the suggestion, referencing the paper of Beliver, 2012. Please check the revised text in lines 138-144.
- The mentioned paper such as Yan et al., (Line 124) and most of them, the author talks about the discoveries without mentioning the techniques used by the researchers.
A: We thank the reviewer for the suggestion; we have added the techniques used by authors of the referenced articles; please see the revised text, in section 3.3.
- They don’t mention either the number of metabolites they analyzed in their papers.
A: we thank the reviewer for the suggestion; we have added the number of metabolites; please see the revised text, in section 3.3.
It is important for this reviewer to understand what techniques are used by the reviewed papers and what they analyze to consider the relevance of the discoveries they publish.
- Line 187. There are machines such as quadrupole that analyzes thousands of metabolites. It would be interesting to present a table with techniques and their limitations in detection (resolution). It would be also important to have a list of method of analysis and their utility (PCA, ANNs, etc).
A: We thank the reviewer for the suggestion; the aim of this study was not to analyze deeply the techniques and their limitations; instead, to examine practically in the clinical level the reason why they have not entered the routine clinical practice. On the other hand, we have provided with the suggested table; please see the newly inserted Table 1.
- Finally, when they say that there is no evidence that the metabolomics can not be used for embryo selection they are not taken into account several abstracts and oral presentations of Cabello et al., at ESHRE and ASRM congress at the last two years. I know that they are publication in congresses but I believe that they discoveries are very promising and could be included in the review.
A: We thank the reviewer for the suggestion; although our search was based in Pubmed, in order to comply with the suggestion, and recognize the value of this promising report, we have added it in the references section and text. Please see revised version (lns 292-4).
Reviewer 2 Report
A metabolomic research based on minireview survey for ART is suggested.
If it is a minireview, please indicate it also in the title
INTRODUCTION – FIRST PARAGRAPH
There is a tendency for the identification and utilization of markers, with sufficient sensitivity and specificity that may lead to change of management to improve outcomes [1, https://doi.org/10.1016/j.cels.2021.06.006, https://doi.org/10.1002/pmic.202000318], in all aspects of medicine.
CONCLUSION
Please, add some future aims regarding authors´ research in the topic.
Author Response
A metabolomic research based on minireview survey for ART is suggested.
The authors are grateful for the time and effort in reviewing this manuscript and the constructive comments aiming to improve the overall performance of this manuscript. Please kindly note that revisions are highlighted using the “Track Changes” function throughout the text. Moreover, you will find the reference list altered. Herein follows the point-by-point response.
If it is a minireview, please indicate it also in the title
A: we thank the reviewer for the suggestion; we have changed the title.
INTRODUCTION – FIRST PARAGRAPH
There is a tendency for the identification and utilization of markers, with sufficient sensitivity and specificity that may lead to change of management to improve outcomes [1, https://doi.org/10.1016/j.cels.2021.06.006, https://doi.org/10.1002/pmic.202000318], in all aspects of medicine.
A: we thank the reviewer for the suggestion; the proposed references have been added, except form the second, as it concerns plants.
CONCLUSION
Please, add some future aims regarding authors´ research in the topic.
A: We thank the reviewer for the suggestion; we have added some sentences in the “suggested steps’. Please check the revised text.
Reviewer 3 Report
-line 77: I would suggest to use also other databases (WoS, Scopus, bioRxiv, medRxiv, etc.) because of the "grey literature" consisting of posters, preprints, etc., which could represent not a minoritary quota of ghost papers, still full of important informations.
-line 83: I would have added even these keywords: "metabolomics","metabonomics","cometabolism","metabolites","byproducts". Especially beacuse some of these keywords are related to the resident microbiota, which produces a great quota of co-metabolites.
-line 88: please add a table reporting the number of papers/reviews retrieved form your research (divided by metabo technique), and the different metabolomics techniques/platforms employed in those papers. As you mention later, results could differ not only from the infertility phenotypes, but even from the metabolomics experimental setup.
-line 94: I agree that metabolomics is more informative than other -omics techniques, but it depends uniquely on the number of variables taken into account. If considering that metabolites could derive also from the resident microbiota (PMID 32760681) and the bacterial byproducts (the genus Prevotella is one of the most metabolically active PMID 33375526), I would add a few sentences on this microbiota co-metabolic aspect.
-line 158: I strongly agree with Authors with this affirmation.
-line 187: see my previous comment on adding a table reporting the different metaboloics techniques.
-line 191: I agree that dimensionality reduction (e.g., feature elimination) is a great issue when coping with few OBS and tons of VARs, but I would add also the collineraity issue that is always present when some variables (bacteria, metabolites, genes, etc.) behave in the same manner. Another issue is that very often the metabolites peaks (NMR or GC/MS) that are low or behind or shoulder of other peaks, are the most informative ones (as usually happens with bacterial species): in order to avoid the problem Pareto scaling or FFT transformation could be employed.
Please add a few sentences (or, better, a table) on these very demanding statistical aspects.
-line 198: I would add also the PCoA (principal coordinate analysis), based on dissimilarity/similarity matrices. PCA instead is based on raw data.
-line 230: for the clinical use of metabolomics (or, in general, for 'omics techniques) is even more important the time in providing results, because clinicians are interested in the personalized medicine approach, which contrasts with the massive NGS sequencing or massive metabolomics. Thus, the bioinformatician/biostatistician would need to provide interpretable results at an affordable time. Please discuss this important aspect.
-line 253: I strongly agree.
Author Response
The authors are grateful for the time and effort in reviewing this manuscript and the constructive comments aiming to improve the overall performance of this manuscript. Please kindly note that revisions are highlighted using the “Track Changes” function throughout the text. Moreover, you will find the reference list altered. Herein follows the point-by-point response.
-line 77: I would suggest to use also other databases (WoS, Scopus, bioRxiv, medRxiv, etc.) because of the "grey literature" consisting of posters, preprints, etc., which could represent not a minoritary quota of ghost papers, still full of important informations.
A: we thank the reviewer for the suggestion; the aim of the study was to to investigate the utilization of metabolomics in assisted reproduction and the potential flaws in its use, using only Pubmed as the only article source; we fully acknowledge the fact that there are many interesting papers in the grey literature that could give useful information on the subject; a typical example of a relevant paper is now added, in lines 292-4.
-line 83: I would have added even these keywords: "metabolomics","metabonomics","cometabolism","metabolites","byproducts". Especially beacuse some of these keywords are related to the resident microbiota, which produces a great quota of co-metabolites.
A: we thank the reviewer for the suggestion; we have added the suggested keywords in lines 89-90.
-line 88: please add a table reporting the number of papers/reviews retrieved form your research (divided by metabo technique), and the different metabolomics techniques/platforms employed in those papers. As you mention later, results could differ not only from the infertility phenotypes, but even from the metabolomics experimental setup.
A: We thank the reviewer for the suggestion. A relevant table summarizing the different metabolomics techniques employed in included papers has been added (Table 1, lines 298-392).
-line 94: I agree that metabolomics is more informative than other -omics techniques, but it depends uniquely on the number of variables taken into account. If considering that metabolites could derive also from the resident microbiota (PMID 32760681) and the bacterial byproducts (the genus Prevotella is one of the most metabolically active PMID 33375526), I would add a few sentences on this microbiota co-metabolic aspect.
A: We thank the reviewer for the suggestion. We have added a couple of lines; please see the revised text (lns 150-4).
-line 158: I strongly agree with Authors with this affirmation.
-line 187: see my previous comment on adding a table reporting the different metaboloics techniques.
A: A relevant table summarizing the different metabolomics techniques employed in included papers has been added.
-line 191: I agree that dimensionality reduction (e.g., feature elimination) is a great issue when coping with few OBS and tons of VARs, but I would add also the collineraity issue that is always present when some variables (bacteria, metabolites, genes, etc.) behave in the same manner. Another issue is that very often the metabolites peaks (NMR or GC/MS) that are low or behind or shoulder of other peaks, are the most informative ones (as usually happens with bacterial species): in order to avoid the problem Pareto scaling or FFT transformation could be employed.
Please add a few sentences (or, better, a table) on these very demanding statistical aspects.
A: A: we thank the reviewer for the suggestion; we have addressed these issues adding 3 sentences and the relevant references. Please check lines 464-73.
-line 198: I would add also the PCoA (principal coordinate analysis), based on dissimilarity/similarity matrices. PCA instead is based on raw data.
A: we thank the reviewer for the suggestion; we have added a short description and a reference in the revised version on PCoA. Please check lines 454-7.
-line 230: for the clinical use of metabolomics (or, in general, for 'omics techniques) is even more important the time in providing results, because clinicians are interested in the personalized medicine approach, which contrasts with the massive NGS sequencing or massive metabolomics. Thus, the bioinformatician/biostatistician would need to provide interpretable results at an affordable time. Please discuss this important aspect.
A: We thank the reviewer for the suggestion; sentences discussing this important aspect have been added in two parts of the paper; please check the revised text.
-line 253: I strongly agree.
Round 2
Reviewer 2 Report
Authors accomplished given remarks